# Consumer Likelihood to Seek Information on OTC Medicines

**DOI:** 10.3390/pharmacy11040128

**Published:** 2023-08-11

**Authors:** Jeffrey G. Taylor, Oluwasola S. Ayosanmi, Sujit S. Sansgiry, Jason P. Perepelkin

**Affiliations:** 1College of Pharmacy and Nutrition, University of Saskatchewan, Saskatoon, SK S7N 5E5, Canada; osa355@mail.usask.ca (O.S.A.); jason.perepelkin@usask.ca (J.P.P.); 2College of Pharmacy, University of Houston, Houston, TX 77204, USA; ssansgiry@uh.edu

**Keywords:** OTC medicines, information demand, consumer behavior, familiarity, self-medication

## Abstract

There is concern as to whether the public use OTC (over-the-counter) medicines with due diligence. The objective was to quantify the likelihood and extent people would seek information on OTC medicines in relation to 10 non-medicine products as a surrogate of the importance consumers place on them. Citizens of one Canadian province estimated the likelihood and extent (scale of 1 to 10) they would search for information when considering a purchase. The survey had two lists—a MIXED products list (5 OTC medicine categories and 10 non-medicine products) and an OTC MEDICINES list (15 categories). Five hundred and seventy-five surveys were obtained (response rate 19.2 percent). The average age was 63.0 years and 61.6 percent were female. The mean search likelihood for the 15 products on the MIXED list ranged from 2.2 to 7.4. There was more intention to search for information involving OTC medicines (mean = 5.0) than non-medicine products (mean = 4.1). There was a weak positive correlation in search likelihood relative to OTC medicine familiarity. This study revealed that the likelihood of searching for information prior to purchase is not particularly robust. With a plethora of information currently available to consumers, motivation to access it is what may need attention.

## 1. Introduction

For any minor ailment, medicines available without a prescription are an important aspect of care. The public use them often and are usually satisfied with the results due to their effectiveness and low risk. Achieving such outcomes means users will have to know what specific agent they need and how to use it appropriately.

As more medicines become available without prescription, there is concern as to whether the public will use such medicines with the due diligence they require. Choosing OTC medicines can be a confusing endeavor, be it due to the number of products available [1] or the ability to correctly interpret symptoms [2]. Sansgiry noted how product line extensions can worsen the problem [3]. For a survey of Canadians, one-third stated they definitely had purchased a wrong medicine [4]. A further third wondered if they also had made the same mistake. Part of this issue is the readability of various information sources, including the packaging [5]. At times, the public may need to ask a pharmacist or doctor for help.

Reports describe how the public seeks health information [6,7]. Specifically for pharmacies, the following have been identified: information sources, what types of information patients feel is important, and the role of labels [5,8,9,10,11,12]. It has been found that older adults take almost twice as much time to view OTC-related information but are more organized in their searches [13]. Regarding broader search dynamics, products available in the retail world have been described as either low or high involvement depending on the amount of effort consumers expend during purchases [14,15]. OTC medicines have been shown to have moderately high involvement [16] but also have been described as having low involvement [17,18]. Others have noted that when consumers are ‘involved’ in the purchase they are more likely to understand label information [19], leading to better decisions [20,21].

Reports have often considered OTC medicines as a broad monolithic group. But the propensity to seek out information likely varies across different agents and treatment conditions. For example, in a survey of 300 subjects, Taylor et al. assessed the likelihood of seeking information from four sources (Internet, friends/family, pharmacists, and doctors) from a repertoire of 21 clinical situations [8]. The public had a higher likelihood of seeking physician advice when a child had diarrhea than when it involved adults.

Package information is critical to this process. In most countries, a medicine that cannot be taken safely and effectively via the information on (and in) the package is likely to remain on prescription. Accordingly, there has been much interest in the readability (and comprehension) of product labels and the propensity of the public to read them [22,23,24,25,26,27].

Information on OTC medicines and the conditions they treat is widely available. While its existence may not be an issue, the willingness to search for (and utilize) it continues to be of concern. The objective of the study is to describe the likelihood and extent information will be sought for a collection of common retail products, both medicines and non-medicines, if they were to be purchased by consumers. The approach considered OTC medicines as smaller categories rather than one all-encompassing group. Comparing OTC medicines to non-medicine entities may act as a surrogate of the importance consumers place on such agents.

## 2. Methods

Data were gathered via an online survey of residents in one Canadian province. For a location of just over 1.1 million people, 384 responses were needed for a margin of error of ±5 percent [28]. Assuming a response rate of 20 percent, the survey link was sent to 3000 provincial residents in December 2021. Data collection ended four weeks after commencement. The university’s survey service that was in charge of conducting it de-identified the data and transferred it to the research team. The study was approved by the review board of the University of Saskatchewan in October 2021 (BEH #2812).

The source of the subjects was a volunteer citizens panel of Saskatchewan residents over 18 years. Citizen and patient panels have been used for access to various subjects [29,30], including those involving OTC medicines [31,32]. Previously, a mail survey was conducted in this province on a similar topic [11] but the changing dynamics of phonebook addresses and rising costs precluded that approach [33,34]. Subjects were chosen randomly from the panel.

Two lists were created for the survey. The first had five OTC categories and 10 common non-medicine retail products (MIXED list). Televisions and socks were presented first to responders (in that order), which was carried out purposely to act as polar boundaries relative to the search likelihood (and extent). Thereafter, products were randomly presented to responders to help reduce response order bias.

The second list comprised 15 OTC medicine categories. Four were carry-overs from the first list (headache, heartburn, cough syrup, and multivitamins) to act as a gauge of internal consistency. The order presented was also conducted at random for this list.

The products chosen were based on previous work [11,35,36,37] and researcher discretion. Fifteen was set as the limit, while other surveys have listed seven or eight [5,22]. In addition, Roper categorized agents for constipation and diarrhea into one category but they were separated here.

Regarding scale length to assess the attributes of each list, Huston used a 7-point scale to determine the likelihood of patients seeking information from pharmacists regarding new chronic medications [30]. A 10-point scale has been used to evaluate a decision-support system to improve the safe use of OTC medicines [38]. For the current study, scales quantified the likelihood and extent of information-searching during product purchases using 10-point scales, with verbal anchors at the poles: *not likely* (1) to *very likely* (10) and *not extensive* (1) to *very extensive* (10).

A third scale was created to quantify familiarity with the products on both lists also using a 10-point scale (*not familiar* (1) to *very familiar* (10)). In one report used for guidance, Droege et al. quantified responder familiarity with OTC medicines relative to risk perceptions [39].

The phrasing used to guide responders for each parameter of the MIXED list was as follows:

### 2.1. Search Likelihood

The following is a list of 15 common products. You likely own or have used some of them. Either way, for this section please imagine you are about to purchase each one (as something new or as a replacement).

We would like to know how likely it is that you would search for information about the item to make a decision. Searching for information can include a range of activities—*simply reading a package (if applicable), asking a friend, brochures, going online, or even asking an expert for help.*

On a scale of 1 to 10, a higher number means MORE likelihood that you would seek out information. Doing so would be important to you before deciding; you would be motivated to start looking. Low numbers mean you would NOT be very inclined to start looking.

### 2.2. Search Extent

If you did feel a need to start looking for information for these same products, how extensive would such a search be? This could vary substantially. A person might use one source for a little information or MANY sources to obtain a lot of information. Searching for information can include a range of activities—*simply reading a package (if applicable), asking a friend, brochures, going online, or even asking an expert for help.* We would like to know how extensive your search for information would be? On a scale of 1 to 10, a higher number generally means more sources would be used. Or it could also mean using one source extensively. A low number means you really would not have intentions to look very far.

### 2.3. Product Familiarity

This section looks at your general familiarity with these items. By that we mean how often you have used them and your overall experience with other products like it.

However, *familiarity* goes beyond simply owning or using one. Considering TVs as an example, the focus includes TVs in general, not just your TV. How do the various brands differ—*picture resolution, internet access, sound quality, etc.?* So, while a person will know some things about their own TV, they may not be too familiar with how TVs differ across all those traits. Conversely, if you just bought a new TV and looked a lot at different models (called *comparison shopping*) before deciding, then you would be more familiar with them than someone who did not do all that.

With that explanation, we would like to know: how familiar are you with each type of product listed here? On a scale of 1 to 10, a higher number means you are familiar with that product category. Low numbers mean you are not very familiar. You may not have owned (or used) it nor performed much comparison shopping with other products like it.

Regarding familiarity, wording and examples were then changed when responders addressed the list of 15 OTC medicines. All mention of televisions (as the example product) and their differences were removed and clinical phraseology (*nausea* as the example OTC medicine category, medicine traits like *product effectiveness* and *side effects*) replaced them.

Product familiarity was originally configured to capture the actual frequency of product use, such as *once a year, once a month, once a week,* etc. However, this seemed less relevant for items on the first list (like a television or a pair of socks). To rectify this and have consistency across both lists, a decision was made to capture impressions of familiarity instead. It was felt that a user may still consider themselves to be ‘familiar’ with an agent even if used less often: using ibuprofen *four times a month* could lead to a level of familiarity similar to that of a multivitamin used *once a day*. A focus on impressions of familiarity also allowed for better comparison between agents. Irrespective of that debate, the measure of familiarity was not a key outcome. Rather, this information was used as a counterpoise to search likelihood, where lower familiarity might lead to more information searching (and vice versa).

During pilot testing, 100 different people from the citizens list were invited to complete the survey and make suggestions, with 14 doing so. Another 20 purposefully-selected people provided in-depth qualitative feedback (word/phrase difficulties, construct issues, etc.). In both cases, minor changes were made. Three experts in consumer behavior and OTC medicines evaluated the scale validity and other facets that have been previously published [40]. The test–retest reliability was measured by manually quantifying the degree of change along the 10-point scales, conducted with a subset of the sample that was asked to complete it twice. There was no tracking system for non-responders and thus no data on their demographic characteristics. However, early responders were compared to late responders as a proxy measure of non-responders. Data were analyzed descriptively, with *t*-tests for mean differences and Pearson r for correlations using SPSS software.

## 3. Results

Eight people were unable to successfully access the survey online. A total of 575 responses were obtained via 3000 contacts for a response rate of 19.2 percent. The average age was 63.0 years and the majority (61.6 percent) were female (Table 1). Most (54.8 percent) had a university education, 85.8 percent had no children at home, 41.3 percent considered themselves in very good health, and 53.7 percent lived in larger cities.

### 3.1. Information Search Likelihood—15 Mixed Products

Table 2 shows the mean search likelihood for the 15 MIXED products. The product with the highest likelihood was a television (7.4) while the lowest ratings were for paper towels and garbage bags (2.2). The five OTC products had means ranging from 4.8 to 5.3; OTC heartburn medicine had the highest rating (5.3) followed relatively closely by the other four agents. Comparatively, bike helmets and coffee makers had means of 6.0 and 5.7 which were 0.7 and 0.4 units (of search likelihood) higher than the highest-rated OTC medicine. However, the mean of the average scores of all five OTC products was 5.0 while that for the 10 non-medicine products was 4.1, a difference that was statistically significant (t = −1.7, *p* < 0.05).

### 3.2. Information Search Likelihood—15 OTC Medicines

The means for the 15 OTC medicines ranged from 5.1 to 7.2 (Table 3). Eye drops (infection) rated the highest (7.2) followed by fever medicine for a child (6.7). Multivitamins had a mean of 5.1 which was consistent for this same product on the MIXED products list (a measure of internal consistency).

### 3.3. Information Search Extent—15 Mixed Products

The means ranged from 1.6 to 6.6 (Table 2). Televisions had the highest rating while socks, paper towels, and garbage bags had the lowest (1.6). The five OTC products had means from 3.9 to 4.3, again with OTC heartburn medicine as the highest. Overall, there was almost a one-unit mean difference between the OTC product group (4.1) and the 10 non-medicine products (3.3) which was statistically significant (t = −1.6, *p* < 0.05).

### 3.4. Information Search Extent—15 OTC Medicines

Table 3 shows the mean extent of information searching for the 15 OTC categories. Drops for an eye infection were rated highest (6.0) followed by fever medicine for a child (5.9), children’s cough syrup (5.8), and antihistamines (5.3).

### 3.5. Product Familiarity—15 Mixed Products

Across the spectrum of medicine and non-medicine entities, respondents were most familiar with OTC headache medicines (5.8) and least familiar with bike helmets (3.3) (Table 4). Socks and shampoo also rated high for familiarity.

### 3.6. Product Familiarity—15 OTC Medicines

Table 5 has the averages for familiarity for the 15 OTC medicines, from a low of 2.6 (Athlete’s foot cream) up to 6.4 (headache medicine). Medicines commonly given to children (fever medicine (3.5), cough syrup (3.3), and diaper rash cream (3.0)) were not especially familiar to the subjects in this sample.

### 3.7. Search Likelihood versus Product Familiarity

Drops for an eye infection (search likelihood of 7.2), fever medicine for a child (6.7), and cough syrup for a child (6.6) had familiarity values of 3.9, 3.5, and 3.3, respectively. For this example of three agents, as familiarity dropped so did the search likelihood, which was not expected. For all agents, there was a weak positive correlation between the search likelihood and product familiarity (r = 0.3, CI = 0.2–0.4).

### 3.8. Scale Reliability

The test–retest reliability was measured by manually quantifying the degree of change along the 10-point scales from time 1 (first completion) to time 2 (several weeks later) for all three measures for both lists. Responders were selected for this by selecting every 10th person who had completed a document until reaching 20 candidates. For the 10-point scales, the majority of scoring changes seen were: no change in the first and second response, one unit of change (for example, scoring a four for time 1 and then a five for time 2), or two units of change.

‘Early’ responders were those who responded to the survey within the first week of the survey (6–12 December 2021) while ‘late’ responders did so in the last week at survey closure. Early responders and late responders were mostly female (62.0 percent and 86.7 percent, respectively). Almost 72 percent of the early responders were elderly while middle-aged respondents made up 50 percent of late responders. A majority (51.2 percent) of early responders resided in the larger cities, whereas 40.0 percent of late responders resided in either the large or smaller cities. Those with a college degree were similar in proportions. Early and late responders both reported good health (87.5 percent and 81.2 percent, respectively). Similarly, most early and late responders had no children living with them (87.3 percent and 81.2 percent).

## 4. Discussion

Once a person decides to manage an ailment, a popular option will be an OTC medicine. Appropriate use depends on the quality of information available as well as the abilities of individuals to utilize it. The extent to which those tenets are met, unfortunately, continues to foster debate. Package information is critical to this process. Accordingly, there has been interest in its’ usefulness and the propensity of the public to read it [5,12,22,23].

As an extension to the current understanding of information-seeking behavior, the current report describes the likelihood and extent information will be sought for common products, both medicine and non-medicine. The approach is somewhat unique in that it considers OTC medicines as sub-groups rather than an all-encompassing group. It had respondents do this relative to the various sources that would be available to them, be that a pharmacist, the Internet, or a friend. Comparing OTC medicines to non-medicine products may reflect the importance the public bestows onto these agents.

For the list of 15 medicine and non-medicine products, televisions ranked highest as the item consumers would most likely seek information prior to a purchase. This was expected and was by design. They are high involvement products and were included to create a ceiling effect; it was predicted that other products would fall below them (as was the case). Socks represented the opposite—items most people likely do not devote much attention to. With those poles in place, the data allow one to see the extent to which OTC medicines gravitate to either pole.

Headache medicine garnered a value of 4.8, falling below a coffee maker at 5.7 and only 1.0 point above shampoo. This may call into question whether consumers indeed consider OTC products as ‘medicines’ or simply another group of goods available to the public. There is some insight for this, including reports that have categorized OTC medicines as low involvement purchases [17,18]. If these values translate into actual medicine-taking behavior, it could be problematic. A complicating matter for this current study, though, is that there is little understanding as to what level of search likelihood might be deemed more appropriate—how many points on the scale should a cough syrup be above shampoo?

For headache medicines (and all others), the premise of this report was the degree of searching that might take place for a future purchase, not what was performed at first purchase. Given that such medicines were the most familiar of any product on the list, it is entirely possible that users had already performed their homework. They were perhaps clear on whether acetaminophen is the better choice over naproxen or ibuprofen (or vice versa), whereby the next purchase did not call for any further deliberation on the matter. Still, the evidence here suggests consumers view OTC medicines as somewhat on par to an array of non-medicine items. As a group of five products, there was a statistically significant greater level of search likelihood compared to the 10 non-medicine products but that difference was not inherently large.

It might seem unproductive to engage in a discussion involving medicines and any comparisons to non-medicine items. Surely, it is highly unlikely that the public equates the dynamics of selecting a heartburn medicine to that of buying a bike helmet or paper towels. There is also data on the extent to which medicine labels are read during first-time selection. Knowing this may attenuate some of the concern raised here, reiterating that users may already have performed due diligence prior to being asked their intentions. As an example, the following statement was asked of British citizens: *when I have a minor ailment, the first thing I do is try and find some relevant health information myself*. Eighty percent of responders agreed or strongly agreed [41]. An industry report noted that 91 percent of Canadians claim to read the label carefully before using a product for the first time [25] and appeared to be satisfied with this information [42]. A majority of Americans also claimed to read at least parts of the package [12]. Of those saying they would re-read labels, 77 percent noted this would happen when giving the agent to a child or if they had not read the label for some time (70 percent). In a survey conducted in the same province as the current report, the majority (86.5 percent) had received medicine-related advice from a pharmacist in the past [11].

There are less encouraging results [5]. Only 40 percent of Canadians claimed to read about active ingredients and even smaller numbers read about dosages (34 percent) and directions for use (18 percent) when buying a product for the first time [43]. There is also concern that a number of people have problems understanding such information [44]. Over half of the responders in a Canadian survey said they found labels of OTC medicines to be always (13 percent) or sometimes (49 percent) difficult to understand [5]. Market researchers have considered OTC medicines and, while Gore et al. found consumers exhibited moderately high involvement when making purchases [16], others found little support for that level [17,18]. Many patients could not think of an OTC category that was important to them and those that could did not indicate a high level of involvement in the number of sources consulted or any perceived risk [17].

The third aspect examined was product familiarity relative to information search likelihood. Familiarity with a product implies, but does not guarantee, that the person has some knowledge or experience with it based on continued use. Early on, searching for information will take place until it is no longer felt to be needed. It, therefore, is likely that when familiarity is low, search likelihood rises. One product (drops for an eye infection) rated high on search likelihood (7.2) and relatively low for familiarity (3.9). This is an example of what was expected with the results—not knowing much about an agent would lead to greater efforts to seek information. Overall, though, there was an unexpected weak positive correlation between search likelihood and product familiarity.

Respondents appeared to be reasonably familiar with the consumer products presented to them, except for bike helmets, red wine, and thermometers. They were most familiar with headache medicine. Specific to the list of OTC medicines, since almost two-thirds of respondents were elderly and had no children at home, it was not surprising they were unfamiliar with ones applicable to pediatric use. However, the low rating for laxatives was somewhat of a surprise since their use generally rises with an elderly audience. This speaks to the limitations of the sample that was surveyed.

### 4.1. Future Research

Work should continue to determine the information needs of people who use OTC medicines such as when they want it and in what format to best digest it.

### 4.2. Practical Implications

Pharmacists and other healthcare providers should continue to be vigilant on whether users have the best information to appropriately select and use such medicines.

### 4.3. Study Limitations

Limitations are inherent to survey research: the distilling of complex human behavior into numerical constructs. For instance, it is not known whether the search likelihood scales actually reflected a propensity to seek out information nor how such propensity changes over the scale gradations. Based on a 10-point scale, one might speculate that each increment was a 10 percent difference but there is no valid basis for such an assumption and it would be extremely difficult to manifest that as (for example) the number of sources utilized or the hours spent searching. Responders may also have been unable to differentiate between the likelihood of engaging in a search and the extent of that search.

There was a risk of response bias during data recovery although altering the order of products presented to responders attempted to attenuate that concern.

The sampling frame should not be considered as reflective of the population under study given that participants were obtained from a citizens panel of volunteers. This was known before proceeding with the study. The sample eventually reflected data of those more educated (as has been seen with another panel [32]), were older, and no longer had kids at home.

## 5. Conclusions

Given the potential for OTC medicines to help resolve symptoms but also to do harm, care must be undertaken when deciding to use one. The evidence suggests the public is more likely to seek information on OTC medicines than mundane entities such as paper towels and body lotion. But those differences were not particularly notable. The fact that a headache medicine garnered a value only one unit above shampoo is quite concerning, regardless of any attenuating effect of familiarity. While there is a plethora of information available to consumers on such medicines, motivating them to access it may continue to need attention.

## Figures and Tables

**Table 1 pharmacy-11-00128-t001:** Sociodemographic Characteristics of Participants.

Characteristic	N	Category	Frequency	Percent
Age (years)	554	Under 20	1	0.2
		20–29	3	0.5
		30–39	22	3.9
		40–49	55	9.9
		50–59	93	16.8
		60–69	218	39.4
		70–79	130	23.5
		80–89	32	5.8
Gender	563	Female	347	61.6
		Male	213	37.8
		Other	3	0.6
Education	564	Some high school	10	1.8
		High-school graduate	57	10.1
		Trade/technical school	90	16.0
		Some college/university	98	17.3
		College or university graduate	309	54.8
Number of household children up to 17 years	565	None	485	85.8
		One	29	5.1
		Two	34	6.0
		Three	12	2.1
		Four	4	0.7
		More than four	1	0.3
Health status	564	Excellent	58	10.3
		Very good	233	41.3
		Good	197	34.9
		Fair	62	11.0
		Poor	14	2.5
Place of residence	556	Large city	299	53.7
		Medium city	34	6.2
		Small town	223	40.1

**Table 2 pharmacy-11-00128-t002:** Search Likelihood and Extent—15 Mixed Products.

Product	Search Likelihood		Search Extent	
	N	Mean (Sd)	N	Mean (Sd)
Television	567	7.4 (3.1)	568	6.6 (2.9)
Pair of socks	552	2.3 (2.1)	553	1.6 (1.4)
Paper towels	565	2.2 (1.9)	562	1.6 (1.4)
Headache medicine	568	4.8 (3.0)	564	4.1 (2.8)
Shampoo	568	3.8 (2.7)	568	2.6 (2.1)
Coffee maker	568	5.7 (3.1)	565	4.6 (2.8)
Sunglasses	564	3.9 (2.9)	560	3.3 (2.5)
Garbage bags	566	2.2 (2.0)	564	1.6 (1.4)
Cough syrup	566	5.2 (2.9)	567	4.2 (2.7)
Bike helmet	567	6.0 (3.2)	565	5.0 (3.1)
Red wine	566	3.9 (2.8)	564	3.0 (2.3)
Thermometer (for fevers)	566	4.8 (2.9)	566	3.9 (2.7)
Multivitamin	559	5.1 (3.0)	559	4.0 (2.7)
Body lotion	562	3.5 (2.6)	567	2.6 (1.9)
Heartburn medicine	564	5.3 (2.9)	562	4.3 (2.8)

**Table 3 pharmacy-11-00128-t003:** Search Likelihood and Extent—15 OTC Medicines.

Medicine	Search Likelihood		Search Extent	
	N	Mean (Sd)	N	Mean (Sd)
Head cold medicine	572	5.8 (2.9)	572	4.7 (2.7)
Laxative	569	5.2 (3.1)	570	4.3 (2.7)
Multivitamin	569	5.1 (2.8)	571	4.2 (2.8)
Antihistamine for allergies	567	6.4 (2.9)	572	5.3 (2.8)
Athlete’s foot cream	570	5.3 (3.0)	571	4.0 (2.6)
Diaper rash cream for an infant	568	5.3 (3.3)	571	4.5 (2.9)
Fever medicine for a child	568	6.7 (3.3)	568	5.9 (3.2)
Diarrhea medicine	570	5.5 (2.9)	566	4.5 (2.7)
Low back pain tablet	569	6.3 (2.9)	570	5.2 (2.8)
Cough syrup for a child	569	6.6 (3.3)	570	5.8 (3.1)
Cold sore ointment	568	5.5 (3.0)	571	4.4 (2.7)
Drops for an eye infection	571	7.2 (2.9)	572	6.0 (2.9)
Headache medicine	568	5.3 (2.9)	571	4.5 (2.8)
Cough syrup	568	5.6 (2.9)	570	4.6 (2.6)
Heartburn medicine	566	5.6 (2.9)	569	4.8 (2.8)

**Table 4 pharmacy-11-00128-t004:** Product Familiarity—15 Mixed Products.

Product	Product Familiarity	
	N	Mean (Sd)
Television	570	5.1 (2.7)
Pair of socks	564	5.2 (3.2)
Paper towels	567	4.9 (3.1)
Headache medicine	566	5.8 (2.7)
Shampoo	569	5.2 (2.8)
Coffee maker	567	4.8 (2.7)
Sunglasses	566	4.1 (2.7)
Garbage bags	564	4.8 (3.1)
Cough syrup	565	4.6 (2.5)
Bike helmet	566	3.3 (2.5)
Red wine	566	3.9 (2.8)
Thermometer (for fevers)	564	3.8 (2.5)
Multivitamin	565	4.8 (2.6)
Body lotion	567	4.4 (2.7)
Heartburn medicine	565	4.5 (2.8)

**Table 5 pharmacy-11-00128-t005:** Product Familiarity—15 OTC Medicines.

Medicine	Product Familiarity	
	N	Mean (Sd)
Head cold medicine	571	5.3 (2.6)
Laxative	568	3.8 (2.7)
Multivitamin	568	4.9 (2.7)
Antihistamine for allergies	569	5.1 (2.8)
Athlete’s foot cream	568	2.6 (2.2)
Diaper rash cream for an infant	569	3.0 (2.6)
Fever medicine for a child	567	3.5 (2.8)
Diarrhea medicine	571	3.9 (2.6)
Low back pain tablet	568	4.8 (2.9)
Cough syrup for a child	569	3.3 (2.6)
Cold sore ointment	569	3.6 (2.8)
Drops for an eye infection	568	3.9 (2.7)
Headache medicine	570	6.4 (2.6)
Cough syrup	567	4.8 (2.5)
Heartburn medicine	569	4.8 (2.9)

## Data Availability

The data presented in this report are available on request from the corresponding author.

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
