# Peer review of "Consumer Likelihood to Seek Information on OTC Medicines"

_pharmacy, 2023, doi:10.3390/pharmacy11040128_

Round 1

Reviewer 1 Report

 Abstract should define abbreviation at first use, and also proof read for sentence structure, some are very short and some end abruptly.

Introduction is well written but needs to consider impact of policy on packaging and labelling. Also, impact on marketing legislation and how this differs in different geographies. 

Line 67 - 5 percents, whats the 28?

why did you assume a response rate of 20% - ref?

why did you choose these two groups? why 5 and why 10?

why tv and socks? confusing

Line 88 - roper, no ref?

line 98 - you have an extra bracket 

how did you actually invite participants ? online, email, phone, letter

how did you fill the survey, online or paper

data protection?

exclusion criteria?

table 2 is not outlined well in the methods - maybe explain more earlier 

why were specific medicines not mentioned?

Is well written overall but needs more depth and linking back to policy and legislation

Good, but a lot of abrupt sentences that are missing the "so what" element 

Author Response

Thank you, reviewers!

My responses in red below. 

Abstract should define abbreviation at first use,

Done for "OTC".

and also proof read for sentence structure, some are very short and some end abruptly.

I will assume that comment applies to these 2 sentences --

     Five hundred and seventy-five surveys were obtained (response rate 19.2
     percent). Average age was 63.0 years and 61.6 percent were female. 

I could add an "and" and splice the 2 together, but will leave that to the discretion of the Editors. With limited word counts, I think sentences should be short. 

Introduction is well written but needs to consider impact of policy on packaging and labelling. Also, impact on marketing legislation and how this differs in different geographies. 

I raise that important issue (and impact) with this passage in the INTRODUCTION (lines 55-56) -- 

     Accordingly, there has been much interest in the readability (and
     comprehension) of product labels and the propensity of the public to read it
     [22-27].

I could have taken another whole page to discuss all that work, but kept it short. I then state that while a lot of information is out there, including the package, the tendency to seek it will be the focus of the report.

Marketing legislation and different geographies goes far beyond what I am able to do in this report, I would think. In every country, a government body would control what appears on a label, and how products can be marketed via advertising. Again, I don't think I could efficiently enter that literature without adding a lot more content, and I am not sure it would add to the work here. I am identifying the potential problem here, as opposed to having done research at this juncture to rectify it.

Later on lines 257 to 259, I again make the links to labelled information.

Line 67 - 5 percents, whats the 28?

That is standard phraseology for surveys -- margin of error. Regarding the "28]", so sorry, the front bracket got left off. That is the reference used for the sample size -- [28].

why did you assume a response rate of 20% - ref?

There is no reference for that. Based on doing about 10 surveys in this province, that seemed like a reasonable goal to hope for from past experience. I have seen up to 57%, but also as low as single digits. While I could cite some unrelated surveys here, nothing I could add would be better than what we know about surveys done locally prior to this one. 

why did you choose these two groups? why 5 and why 10?

The reason for 2 groups was the essence of the actual study -- a look at non-drug entities vs drug entities. Thus, 2 groups were needed.

Regarding 5 OTC agents on the first list, and not 10?

We did not need more (ie 10) b/c the 2nd list was the focus on medicines.

The 5 of the first list was to allow for the comparison to the other 10 non-drug agents, the essence of the work.

Should that have been 6? Or 4? Or 8? Perhaps. There was NO precedent at all to help guide this, so we had to start someplace. And if you add more OTC agents on that list, it would stand to reason that you would have to take off a number of non-drugs from the list; we felt 15 was all a responder would want to deal with.

why tv and socks? confusing

On line 81-82, this statement appears to explain that: 

     Televisions and socks were presented first to responders (in that order), done
     purposely to act as polar boundaries relative to search likelihood (and extent). 

In almost every survey I come across about OTC medicine attributes using a Likert scale, responders are simply asked to pick a number on that scale. Then they respond to the other agents on the list. For us, we needed the context of relativity to non-drug entities. So we absolutely needed to establish the polar ends in order to see where the OTCs slotted in? Thus, TVs (likely high end searching) and socks (likely low end searching) provided the theoretical extremes for search likelihood. If we did not do what we did, and lets say posted multivitamins first, would they have picked 6 (whatever) on the scale? Then what happens when they then get to a TV or a bike helmet? This process add relativity to the non-drug entities, and established a potential boundary right at the start. 

I would be more than happy to go into great detail on this, but every editor I have dealt with wants LESS words. 

Lastly, to further explain all this, lines 264 to 270 then re-addresses this issue. 

Line 88 - roper, no ref?

Roper was cited right before -- ref 22 -- and I used the phrase "In addition" to indicate a continuation of that reference's content. 

line 98 - you have an extra bracket 

If this refers to this passage, I think I am actually missing a bracket at the end:

     scale (not familiar (1) to very familiar (10))

how did you actually invite participants ? online, email, phone, letter

Line 66 to 73 lists this out:

     Data was gathered via an online survey of residents in one Canadian
     province.
     The survey link was sent to 3000 provincial residents in December 2021.
     The source of subjects was a volunteer citizens panel of Saskatchewan
     residents.

We had access to a citizens panel of email addresses, done online

how did you fill the survey, online or paper

As above. Stated directly at line 66, then I explained why paper was not done at line 75-77.

data protection?

Of course, we met all the requirements of the Ethics Review Board. Does that have to actually be written up in the report?

exclusion criteria?

We used a volunteer panel with the University's survey research office, as stated on line 73:

     The source of subjects was a volunteer citizens panel of Saskatchewan
      residents over 18 years.

Opinions were being obtained here, so we had no need to exclude anyone based on actual OTC product usage, age, gender, or other factor. With citizens panels, you simply get access to those who volunteered to be on it.

table 2 is not outlined well in the methods - maybe explain more earlier 

I am at a loss on how to address this comment.

Lines 79 to 128 literally covers the approach taken that led to Table 2, the most important table in this report. Those lines explained the derivation of the MIXED products list, the number of items, the order they were presented, the scale length, and the actual wording used for responders on both LIKELIHOOD and EXTENT parameters.

Then further on lines 185 to 193, the findings were presented to further explain the table. 

I just don't see how Table 2 might be interpreted as somewhat of a surprise at this juncture. 

why were specific medicines not mentioned?

That is b/c we did not look at specific medicines.

The burden to responders would have been huge if we went with specific medicines like DM syrup, loratadine, PEG 3350, docusate, pseudoephedrine, clotrimazole, ibuprofen, loperamide, bacitracin drops, and so on, rather than categories. The lists would have been longer than 15 for sure.

Categories has been the precedent, and we took that further with some of our classes. For example, as state on lines 89-90, Roper categorized agents for constipation and diarrhea into one category, but they were separated here, a good move. 

But there is a big limitation with even going with categories, as explained on lines lines 282-284 (ibuprofen vs acetaminophen). 

Is well written overall but needs more depth and linking back to policy and legislation

This is the 3rd article from this project, so I don't think the overall premise is lacking depth. But if more material is wanted from the Editors, just let me know.    Regarding linking back to policy and legislation, I am curious as to what is missing? Is this a suggestion to legislate consumers to read package info? Is it a suggestion that since consumers may not look "enough", that all such agents should be placed behind-the-counter? Is is a suggestion to make labels more useable or readable (which I cited). Regarding policy, I alert readers to the idea that consumers may not be giving these agents their due attention, and that more interaction with pharms would be great, but I have no plans to say how to do a better job on that front, in spite of 30 years looking at it. I have very little comfort to enter that foray, even tho I could offer up a lot of opinion. Based on the specific details of this report, I went as far as I felt was acceptable.    Comments on the Quality of English Language

Good, but a lot of abrupt sentences that are missing the "so what" element 

I politely disagree on the abruptness of sentences, and would rather state that it was written concisely, to the point.

Regarding the "so what" element, here are some:

     Lines 262-263: Comparing OTC medicines to non-drug products may reflect
     the importance the public bestows onto these agents. 

     Lines 271-274: Headache medicine garnered a value of 4.8, falling below a
     coffee maker at 5.7 and only 1.0 point above shampoo. This may call into
     question whether consumers consider OTC products as indeed ‘medicines’ or
     simply another group of goods available to the public. 

     Lines 275-276:  If these values translate into actual medicine-taking behavior,
     it could be problematic. 

Respectfully submitted.

Reviewer 2 Report

Dear authors,

OTC use is on the rise and the population is aging in most countries. Thus there are good reasons to ask questions about patients' use of information on OTC medicines. 

The presented study uses a newly elaborated survey and the authors mention prior work (ref 39), but it remains unclear what of the validated instrument has been used in the present survey. The whole development of the survey instrument, including some validation of reliability, is rather short and could be detailed more.

Ideally, a mixed methods research may be more suitable to answer the research question on patients' propensity to investigate OTCs before using them. A theoretical framework, such as the Theory of planned behavior, could prove useful in such a larger project.

As presented, the project presents several limits: little information on any theoretical framework, little information on the validation of the survey instrument and little generalizability of the results. Regarding this last point, a comparison of the characteristics of the survey population to the general population of Saskatchewan might be reassuring. One could compare the survey results for the older part of the study population (50+ or 60+), which is overrepresented in the survey in comparison to all of Saskatchewan, to the province population of these age groups, regarding the proportion of women, educational attainment, etc.

Finally, I found the abstract not sufficiently precise. The term "surrogate" remains unclear, as the sentence of line 16 and the term "robust" on line 18.

The article is easy to read but there is a lack of precision at many places. I point out  some:

1. Please use either "drug" or "medicine" throughout the manuscript - I prefer medicine.

Line 25: a reference would be useful here.

Line 67: a reference for the sample size calculation would be good.

Line 88: which of the before mentioned (?) references is "Roger"?

In the Methods section one usually does not compare the chosen method to those in the literature, unless published methods are used in the reported study. Otherwise such comparisons should be moved to the discussion.

Line 157: I'ld rather say "...to a level of familiarity similar to that....."

Line 252: "smaller entities"...I'ld rather say: subgroups of OTC medicines

Line 254-255: "...these agents". : which agents are meant?

Line 269: there may be one "is" too many or something else is missing

Line 296: "received advice from a pharmacist...." On what did they receive advice?

Author Response

Thank you, reviewer 2.

My responses are in red below. 

OTC use is on the rise and the population is aging in most countries. Thus there are good reasons to ask questions about patients' use of information on OTC medicines. 

I worry about this every time I work at the pharmacy. 

The presented study uses a newly elaborated survey and the authors mention prior work (ref 39), but it remains unclear what of the validated instrument has been used in the present survey. The whole development of the survey instrument, including some validation of reliability, is rather short and could be detailed more.

Reference 39 was an extensive look at the validation and reliability issue. It is easy to get bogged down by the details, so it was set aside for a separate article.

I can add more tho, if so desired by the editors. 

I would argue against that, tho. Validity is not a key issue in this report with the lists utilized.

The instrument was a list of 15 agents (x 2). There can be no real validity component to that, its just a list. There is NO instrument to speak off, as in, conversely if I was looking at attitudes towards OTC medicines. 

The familiarity construct was not a list of attitudinal or belief items to somehow measure familiarity. So, again, the need for "validity" would be a big over-sell here. 

That said, did we pick the "correct" 15 items? I am not sure any amount of "validity testing" would sort that out.

Regarding familiarity with products, I cannot vouch for how accurately it reflected consumer familiarity, but how would one even measure that? On lines 153-159, I discussed the problem we faced, and accepted the perception of consumers. 

I will concede that with the 10 point scales used, I wished I could say they were valid measures of likelihood, extent, and familiarity, but cannot. I did not see any reasonable method before we started, tho, to show evidence of validity, except that it would entail interviewing people about actual product use, then correlate that back to the numbers they picked on the scale. At this juncture of the research, I am completely fine with their perceptions of those constructs, rather than any evidence of congruency to actual reality.

Lastly, I felt this would suffice for this article:

     Three experts in consumer behavior and OTC medicines evaluated scale
     validity and other facets that have been previously published [39].

Ideally, a mixed methods research may be more suitable to answer the research question on patients' propensity to investigate OTCs before using them. A theoretical framework, such as the Theory of planned behavior, could prove useful in such a larger project.

Mixed methods would probably be the best approach for the entire OTC area. But you do what you can, at the time, with the resources you have. 

Our approach was based on a look at several theories, which were cited in the PhD thesis. We went extensively into the marketing world and patient illness behaviour to find those, and to make sure we covered our bases. 

As presented, the project presents several limits: little information on any theoretical framework,

As above. We have all that, but for this finite slice of the work (it is the 3rd article emanating from our main work), we wanted to be more focused and concise. 

little information on the validation of the survey instrument

As above. This was not a survey instrument in the usual sense, it was simply 2 lists.

Conversely, for our work on attitudes and illness behaviour for the first report we published, we went into great detail on that. 

and little generalizability of the results.

I felt we were pretty clear:

     The sampling frame should not be considered as reflective of the population
     under study, given that participants were obtained from a citizens panel of
     volunteers. This was known before proceeding with the study. The sample
     eventually reflected data of those more educated (as has been seen with
     another panel [32]), were older, and no longer had kids at home.

Regarding this last point, a comparison of the characteristics of the survey population to the general population of Saskatchewan might be reassuring. One could compare the survey results for the older part of the study population (50+ or 60+), which is overrepresented in the survey in comparison to all of Saskatchewan, to the province population of these age groups, regarding the proportion of women, educational attainment, etc.

We did compare our demographics to that of the province for the main report, and took those main conclusions into this report for the limitations section.

Those results were not "reassuring" tho. We were quite concerned that our results were not representative of the general population, which we clearly state. 

Lines 177 to 180 spell out what we found. We could add a lot about how that differed from the population as a whole, but we did that for the 2 earlier reports, and did not want to get into repetition. We know seniors were over-represented.

Finally, I found the abstract not sufficiently precise. The term "surrogate" remains unclear, as the sentence of line 16 and the term "robust" on line 18.

Surrogate = a substitute, a proxy measure.  Robust = strong.    I would stick with those words.   Comments on the Quality of English Language

The article is easy to read but there is a lack of precision at many places. I point out  some:

1. Please use either "drug" or "medicine" throughout the manuscript - I prefer medicine.

Good point, thank you. Done. 

Line 25: a reference would be useful here.

I felt that was a given, given the massive amount of use of these agents, but will work with the Editors on that, sure. 

Line 67: a reference for the sample size calculation would be good.

That was done -- ref 28. Sorry, the front bracket was missed. 

Line 88: which of the before mentioned (?) references is "Roger"?

Roper -- reference 22. Cited in sentence right before.

In the Methods section one usually does not compare the chosen method to those in the literature, unless published methods are used in the reported study. Otherwise such comparisons should be moved to the discussion.

That is where I originally had that, but for this report, it made much more sense to have those decisions related to Methods right at that juncture. Things got too disconnected when they were put into the Discussion. I am quite happy where they currently lay. 

Line 157: I'ld rather say "...to a level of familiarity similar to that....."

Done. 

Line 252: "smaller entities"...I'ld rather say: subgroups of OTC medicines

Done. 

Line 254-255: "...these agents". : which agents are meant?

"OTC medicines", as mentioned at the start of the sentence. 

Line 269: there may be one "is" too many or something else is missing

Fixed. 

Line 296: "received advice from a pharmacist...." On what did they receive advice?

Fixed.